# Potential Distribution of *Cedrela odorata* L. in Mexico according to Its Optimal Thermal Range for Seed Germination under Different Climate Change Scenarios

**DOI:** 10.3390/plants12010150

**Published:** 2022-12-28

**Authors:** Salvador Sampayo-Maldonado, Cesar A. Ordoñez-Salanueva, Efisio Mattana, Michael Way, Elena Castillo-Lorenzo, Patricia D. Dávila-Aranda, Rafael Lira-Saade, Oswaldo Téllez-Valdés, Norma I. Rodríguez-Arévalo, Cesar M. Flores-Ortiz, Tiziana Ulian

**Affiliations:** 1Plant Physiology Laboratory, Biotechnology and Prototypes Unit (UBIPRO), FES Iztacala, Universidad Nacional Autónoma de Mexico (UNAM), Tlalnepantla 54090, Estado de Mexico, Mexico; 2Royal Botanic Gardens, Kew, Wakehurst, Ardingly, Haywards Heath, West Sussex RH17 6TN, UK; 3Natural Resources Laboratory, Biotechnology and Prototypes Unit (UBIPRO), FES Iztacala, Universidad Nacional Autónoma de Mexico (UNAM), Tlalnepantla 54090, Estado de Mexico, Mexico; 4National Laboratory in Health, FES Iztacala, Universidad Nacional Autónoma de Mexico (UNAM), Tlalnepantla 54090, Estado de Mexico, Mexico

**Keywords:** global change, MaxEnt, distribution model, cardinal temperatures, Spanish cedar, temperature thresholds

## Abstract

*Cedrela odorata* is a native tree of economic importance, as its wood is highly demanded in the international market. In this work, the current and future distributions of *C. odorata* in Mexico under climate change scenarios were analyzed according to their optimal temperature ranges for seed germination. For the present distribution, 256 localities of the species’ presence were obtained from the Global Biodiversity Information Facility (GBIF) database and modelled with MaxEnt. For the potential distribution, the National Center for Atmospheric Research model (CCSM4) was used under conservative and drastic scenarios (RCP2.6 and RCP8.5 Watts/m^2^, respectively) for the intermediate future (2050) and far future (2070). Potential distribution models were built from occurrence data within the optimum germination temperature range of the species. The potential distribution expanded by 5 and 7.8% in the intermediate and far future, respectively, compared with the current distribution. With the increase in temperature, adequate environmental conditions for the species distribution should be met in the central Mexican state of Guanajuato. The states of Chihuahua, Mexico, Morelos, Guerrero, and Durango presented a negative trend in potential distribution. Additionally, in the far future, the state of Chihuahua it is likely to not have adequate conditions for the presence of the species. For the prediction of the models, the precipitation variable during the driest month presented the greatest contribution. When the humidity is not limiting, the thermal climatic variables are the most important ones. Models based on its thermal niche for seed germination allowed for the identification of areas where temperature will positively affect seed germination, which will help maximize the establishment of plant populations and adaptation to different climate change scenarios.

## 1. Introduction

Climate change is going to alter the patterns of rainfall and temperature in various regions of the country. As a result, a higher frequency of extreme meteorological events will affect subtropical and tropical forests, affecting species phenology (e.g., flowering and germination time) [1,2,3,4,5,6,7], and modifying the structure and composition of the tree communities, which will alter migration and its geographical distribution [8,9,10,11,12,13,14,15,16]. Among the factors that shape species distribution, the dispersal of its seeds and the environmental conditions suitable for germination play important roles [17,18,19]. Therefore, it is necessary to consider species optimum germination temperature range in order to identify the new areas in which it will be possible to find the species as well as where the environmental conditions will not allow its germination and establishment in the future.

*Cedrela odorata* L. provides several ecosystem services, among which CO_2_ capture and water balance stand out [20]. The wood of *C. odorata* L., due to its color and texture, is sold as sawn wood and plywood for the manufacture of furniture, for which there is a high demand in the national and international markets. Considering the *C. odorata* wood reserves in natural forests, according to the information generated by the National Forest Inventory, Romo-Lozano et al. [20] estimated that the species represents a financial value of USD 288,328,788 for Mexico. For these reasons, it is a species of major importance in the Mexican forestry industry [21]. However, the intensive extraction of this wood has decreased its populations and areas of distribution. Therefore, the species has been overexploited and threats to this species are exacerbated by its vulnerability to pests [7].

Due to its vulnerable conservation status in natural forests, *C. odorata* is included in Appendix II of the Convention on International Trade in Endangered Species [22]. In addition, it is listed as an endangered species subject to special protection in the Official Mexican Standard 059 (NOM-059-SEMARNAT) [23]. Finally, climate change has already been reported to alter its phenology, interactions, distribution, morphology, and net primary productivity in tropical forests [24].

As an effect of climate change, an increase in temperatures and a decrease in precipitation are forecasted for Mexico [7]. Thus, studies on potential species distribution are needed in order to understand how a warming climate will impact the development of a single species [25]. Several studies have already been carried out to model and predict the future distribution of *C. odorata*. Hernández-Ramos et al. [26] generated an ecological niche model using 19 climate and terrain variables as well as altitude, orientation, slope, and soil type layers, which suggested a reduction in the ecological niche of the species in the country. In addition, Gómez-Díaz et al. [25] estimated the potential climate areas of the distribution of *C. odorata* in the state of Hidalgo, and they obtained conflicting results according to the two applied climate change scenarios. For the American model (GFDL-R30) [27], an expansion in the surface of 3.1 and 4.4% is expected for the years 2020 and 2050, respectively, while for the English model (HadCM3) [27], a contraction in the surface of 0.9% for 2020 and 0.2% for 2050 is expected.

Species distribution models (SDM) have been used to analyze the impact of climate change on the distribution of species, among which the Maximum Entropy model (MaxEnt) stands out for its precision [28]. It feeds on species presence records as well as environmental and climatic variables. However, when physiological variables are incorporated into the model, it becomes stronger and increases in precision. The model performs a maximum likelihood estimation of the relative probability of presence and analyzes the relationship between species distribution dynamics and climate change [17,19,28].

For the study of plant distribution, bioclimatic indices are usually used, which are based on temperature and precipitation data as climatic variables. These indices define the physiological limitations of the species, where the minimum temperature is a factor that can limit the growth and development of plants, especially for those from warm regions such as the species under study [29]. Funes et al. [30] identified temperature as the main regulating factor that triggers germination and maximizes seedling establishment, survival, and regeneration in the understory of forests. Durán-Puga et al. [31] mentioned that faster germination rate confers a competitive advantage in colonizing fragmented habitats. Temperature is an important element of the climate, suitable for analyzing the response of seed germination under different climate change scenarios [32].

González [33] stated that temperature influences the physiological processes of seed germination since it determines both the speed and maximum percent of germination. García-Huidobro et al. [34] concluded that the maximum and minimum temperatures determine the limits within which germination is possible, while the optimum temperature, in agreement with Gilbertson et al. [35], is the temperature at which germination is fastest and seedlings are acquired in the amount least time.

Calzada-López et al. [36] support the use of the cardinal temperatures—in order to find the optimum temperature where the highest amount of germination occurs in the shortest time—as a criterion for the distribution of the species as a direct effect of adaptation to different climate change scenarios [37]. *C. odorata* seeds are reported to be non-dormant [7,38]. Seeds of this species have a wide range of temperatures for germination, ranging from −0.5 ± 0.09 to 53.3 ± 2.1 °C, while the thermal time was 132.74 ± 2.60 °C, which accumulates in a shorter period due to climate change, thus accelerating the germination of seeds in the soil. [7]. It is important to mention that there are no previous studies that take into account optimum germination temperature range in the potential distribution of this species.

According to Adam et al. [39], cardinal temperatures for germination tend to be similar to those of normal vegetative growth of plants, although, in some cases, it may need higher temperatures for flowering. Thus, the environmental conditions where the species is distributed (suitable climate habitat), especially where the temperature is in the range of temperature thresholds for germination, are of paramount importance. In this sense, to consider the optimum temperature range when modelling the potential distribution of *C. odorata* lays the basis for locating areas where the temperature will have a positive effect on seed germination, thus establishing the species in changing climate conditions [40]. The objective of this study was to model the current and future distributions of *C. odorata* according to its thermal niche for seed germination under different climate change scenarios using environmental variables with two representative concentration pathways (RCP) of CO_2_ emissions.

## 2. Results

### 2.1. Current Distribution of C. odorata

Using the georeferenced data from the Global Biodiversity Information Facility (GBIF) and the BioClim database from the period of 1970 to 2000, a map of the current distribution of optimal climate habitat was generated for *C. odorata*. The model was built from species presence data that were within the optimum germination temperature range. The current surface of *C. odorata* with optimal climatic habitat for germination was 228,780.66 km^2^. According to the MaxEnt model, it is distributed from 10 to 28.7 °C annual average temperature, which falls into the suboptimal range for seed germination (Figure 1). According to the jackknife, the variables that contributed most to the model were Bio14 (31.4%), Bio12 (29.6%), and Bio6 (23%).

### 2.2. Potential Distribution of C. odorata

According to the NCAR model (CCSM4) for the intermediate future (2050) under the conservative scenario (RCP2.6 Watts/m^2^), it is predicted that the potential distribution will be of 243,890.85 km^2^ with optimal climate habitat (Figure 2 and Table 1). This represents a 6.6% expansion compared with the current distribution. According to the jackknife, the variables that contributed most to the model were Bio14 (36.8%), Bio6 (28.3%), Bio12 (11.6%), and Bio7 (9.3%).

The states in which the distribution surface with optimal temperatures for germination will expand compared with the current model are Michoacan (219%), Hidalgo (179%), Queretaro (92%), Colima (70%), Jalisco (70%), Sinaloa (50%), Oaxaca (27%), and Veracruz (26%). In contrast, the states whose surface with temperatures for germination of *C. odorata* will contract are Chihuahua (93%), Morelos (84%), Mexico (84%), Guerrero (64%), Puebla (36%), Sonora (30%), San Luis Potosi (28%), Durango (25%), Tamaulipas (18%), and Quintana Roo (14%), compared with the current model (Table 1). The state of Guanajuato, which does not have *C. odorata* populations in the current scenario, in the conservative scenario will present 15.38 km^2^, which will be located in the northeast part of the state in the municipalities of Xichu and Victoria.

For the drastic scenario (RCP8.5 Watts/m^2^), it is predicted that the potential distribution will be of 241,533.21 km^2^ with optimal climate habitat (Figure 2 and Table 1). This represents a 5.57% expansion compared with the current distribution. However, it represents a 1.03% contraction compared with the conservative scenario in the intermediate future. According to the jackknife, the variables that contributed most to the model were Bio14 (43.3%), Bio6 (28.7%), Bio12 (9.7%), and soils (4.3%).

The states in which the distribution surface with optimal temperatures for germination will increase compared with the current model are Hidalgo (134%), Sonora (98%), Jalisco (75%), Sinaloa (47%), Queretaro (34%), Veracruz (28%), Oaxaca (27%), and Chiapas (22%). In the opposite case, the states whose surface with temperatures for germination of *C. odorata* will decrease are Chihuahua (99%), Morelos (91%), Mexico (81%), Guerrero (62%), Durango (48%), Colima (37%), San Luis Potosí (25%), Tamaulipas (25%), and Quintana Roo (18%) compared with the current model (Table 1). In the intermediate future, Guanajuato will present 6.64 km^2^, which will be located in the northeast part of the state in the municipalities of Xichu and Victoria.

For the far future (2070) in the NCAR model (CCSM4) under the conservative scenario (RCP2.6 Watts/m^2^), it is predicted that the potential distribution will be of 239,938.67 km^2^ with optimal climate habitat (Figure 3 and Table 1). This represents a 4.87% expansion compared with the current distribution. According to the jackknife, the variables that contributed most to the model were Bio14 (37.9%), Bio6 (24.5%), Bio12 (18.5%), and Bio7 (5.2%).

The states in which the distribution surface with optimal temperatures for germination will increase compared with the current model are Michoacan (199%), Hidalgo (173%), Queretaro (120%), Jalisco (81%), Sinaloa (48%), Colima (41%), Tabasco (29%), Veracruz (28%), and Chiapas (22%). In the opposite case, the states whose surface with temperatures for germination of *C. odorata* will decrease are Chihuahua (96%), Morelos (92%), Mexico (89%), Sonora (77%), Guerrero (64%), San Luis Potosi (35%), Durango (26%), Quintana Roo (22%), and Tamaulipas (18%) compared with the current model (Table 1). In the intermediate future, Guanajuato will present 65.1 km^2^, which will be located in the northeast part of the state in the municipalities of Xichu and Victoria.

For the drastic scenario (RCP8.5 Watts/m^2^), it is predicted that the potential distribution will be of 246,681.45 km^2^ with optimal climate habitat (Figure 3). This represents a 7.82% expansion compared with the current distribution, representing a 2.81% expansion compared with the conservative scenario for the far future. According to the jackknife, the variables that contributed most to the model were Bio14 (43.8%), Bio6 (25.8%), Bio7 (11.8%), and soils (4.3%).

The states in which the distribution surface with optimal temperatures for germination will expand compared with the current model are Colima (177%), Hidalgo (162%), Sonora (94%), Michoacan (83%), Jalisco (78%), Sinaloa (49%), Puebla (38%), Veracruz (31%), Chiapas (28%), and Oaxaca (19%). In the opposite case, the states whose surface with temperatures for germination of *C. odorata* will decrease are Chihuahua (100%), Mexico (98%), Morelos (91%), Guerrero (55%), Durango (49%), San Luis Potosi (24%), and Tamaulipas (23%) compared with the current model (Table 1). The state of Guanajuato in the intermediate future, will present 2.7 km^2^, which will be located in the northeast part of the state in the municipalities of Xichu and Victoria.

From the current to the potential distribution of *C. odorata*, the states of Chihuahua, Mexico, Morelos, Guerrero, and Durango presented a negative trend in land surface with optimal temperatures for germination. The most extreme case was the state of Chihuahua, which, in the current scenario, shows 433.45 km^2^, but in the intermediate future, it loses more than 93%, and for the far future, no surface is predicted. In the opposite case, the states of Michoacan and Hidalgo will present positive trends in surface with optimal temperature for germination. On the other hand, the models generated predict the distribution in optimal germination temperature range for the state of Guanajuato, which is indicative of the impact of climate change in that region. According to the above, the state of Guanajuato will present favorable conditions for *C. odorata*. Therefore, climate change can have a positive impact on the state’s forest production, but at the same time, it shows the lack of policies and programs to mitigate the negative impacts on the country’s fragile ecosystems.

According to the jackknife method, the variables that alone explain, to a greater extent, the models of the distribution of *C. odorata* were precipitation during the driest period, minimum temperature of the coldest month, and the annual temperature oscillation variable. The soil type variable has information that is not contained in any other variable; its presence is relevant since eliminating it would result in a decrease in the precision for the models. The generated prediction models presented area under the curve (AUC) values of 0.974 ± 0.007, which indicates a good adjustment of the models to the prediction of the areas most likely to find the species based on its temperature thresholds. For each scenario, more than 1500 candidate models were generated; the selected model was the one with an omission rate of 0.05%, a receiver operating characteristic (ROC) of 1.3, and a model complexity (AIC) of 2907.

## 3. Discussion

### 3.1. Current Distribution of C. odorata

The modeling of current and future distributions was based on the statistical principle of maximum entropy, according to Philips et al. [41], allowing predictions considering only records of species presence. For modeling the optimal distribution on the basis of its temperature thresholds, the environmental variables obtained from BioClim were used. The climate variables of BioClim have been used previously in the modeling of potential optimal distribution of forest species from Mexico, including *Pinus chiapensis* (Martínez) Andresen [42], *Abies religiosa* (Kunth) Schltdl and Cham [43], *Pinus leiophylla* Schiede ex Schltdl and Cham [44], *Swietenia macrophylla* King [45,46], and *Cedrela odorata* L. [26]. The current distribution model of *C. odorata* coincides with the distribution of the humid tropical forest biome and part of the seasonally dry tropical forest in Chihuahua, Sonora, Durango, and Sinaloa [47]. They present annual precipitations of more than 1200 mm, with maximum temperatures of 34 to 40 °C during spring and summer (end of March to end of August) and without frost.

The current distribution models obtained in this work coincide with those reported by Hernández-Ramos et al. [26]. The variables that contributed highly to the prediction of the model of current distribution of *C. odorata* by cardinal temperatures were precipitation during the driest period, annual precipitation, and minimum temperature of the coldest month, with 84%. The variables for the ecological niche modeling in this species are in agreement with Hernández-Ramos et al. [26], and they were annual precipitation, precipitation during the driest period, diurnal temperature oscillation, and altitude, with an 88.1% contribution. At the same time, the results are comparable to the potential distribution in Mexico of the optimal climate habitat for *S. macrophylla* (classified in the same family, Meliaceae), in which the variables with the largest contribution were precipitation during the driest month, minimum temperature of the coldest month, temperature seasonality, and annual temperature range; with a contribution of 78.1% [46], while in another study of *S. macrophylla* [45], the variables with the largest contribution were mean temperature during the coldest month, mean temperature, minimum temperature, and precipitation during the growing season, with a contribution of 66.7%. According to Harmann et al. [48], the distribution of forest species is determined mainly by temperature. A recent investigation [49] confirms that the temperature thresholds on the coldest days is the most important environmental variable to determine the distribution of the types of vegetation that exist in the world. This observation confirms the main patterns observed in this and previous studies [29] and highlights the importance of temperature in predicting the distribution of tree species in Mexico.

### 3.2. Potential Distribution of C. odorata

The variables that contributed the most to the models’ prediction of the potential distribution of *C. odorata*, based on its temperature thresholds for seed germination, were precipitation during the driest period, minimum temperature of the coldest month, annual precipitation, and annual temperature oscillation, with 86% in the conservative scenario for the future 2050 and 2070. However, in the model of the future distribution of the species proposed by Hernández-Ramos et al. [26] for 2050, the mean temperature of the warmest quarter, altitude, minimum temperature of the coldest month, and temperature seasonality contributed, with a 92% in the conservative scenario. In turn, Sampayo-Maldonado et al. [46] built a model to predict the distribution of *S. macrophylla* in the conservative scenario for the future 2050 and 2070. The variables that contributed the most were precipitation during the driest month, minimum temperature of the coldest month, temperature seasonality, annual temperature range, and altitude, with 85.5%. In another model for the species [45], for the 2030 in the intermediate scenario (RCP6), the variables that contributed the most were average temperature of the coldest month, minimum average temperature, and growing season precipitation, with 66.7%.

For the drastic scenario in the intermediate future (2050) in this work, the variables precipitation during the driest period, minimum temperature of the coldest month, annual precipitation, and soil type contributed, with 86%, while for the far future (2070), they contributed with 85.7%, which is very similar for the distribution of *S. macrophylla* [46]. Since the thermal variables are the same, they contributed from 80.4 to 83.2% for the construction of the distribution models of this species. Although the annual precipitation variable did not contribute, the annual temperature does an oscillation variable. Therefore, the temperature is a variable that should always be considered [29,49]. According to González [33], if the humidity is not limiting, the temperature is the most determining factor for the germination and distribution of the species. Each species has its own place in space and time that can be represented on a map [50].

According to the NCAR model (CCSM4) with a probability greater than 20%, under the intermediate future (2050) in the conservative and drastic scenarios, the potential distribution will expand 6.6 and 5.57%, respectively, compared with the current distribution. On the other hand, for the far future (2070), the potential distribution will expand 4.87 and 7.82% for the conservative and drastic scenarios, respectively, compared with the current distribution. This is without considering external factors, such as anthropogenic activities that lead to changes in land use [51]. Using the same model (CCSM4), Hernández-Ramos et al. [26] reported a contraction in the distribution of the species by 2050, with a model probability greater than 50%; however, they did not mention the percent of contraction. Another study used the same model (CCSM4) for the potential distribution of *S. macrophylla* [46], and they reported for 2050 an expansion of 18.3% in the conservative scenario and 14.4% in the drastic scenario. For 2070, they reported expansions of 17% and 12.3%, respectively, all with respect to the current distribution, with a probability greater than 20% in the construction of the models. Other authors [45] reported a 60% contraction in the distribution of *S. macrophylla* in the Yucatan Peninsula by 2030. They used a model based on averaging 18 climate models with intermediate greenhouse gas concentrations (RCP6.0 Watts/m^2^), which is a moderate scenario, although the probability for the construction of the model is not mentioned.

According to the species’ distribution models, an expansion in the area is expected in the future with respect to the current distribution, likely related to the rise in minimum temperature. However, when comparing the scenarios in the intermediate future (2050), in the drastic scenario (RCP8.5 Watts/m^2^), there was a contraction of 1.03% compared with the conservative scenario (RCP2.6 Watts/m^2^), which can be explained by the 1.9 °C temperature increase between the scenarios. For the far future (2070), in the drastic scenario (RCP8.5 Watts/m^2^), there was a 2.95% expansion compared with the conservative scenario (RCP2.6 Watts/m^2^), which can be explained by the 2.0 °C temperature increase between the scenarios. Therefore, according to the above, the increase in temperature benefits the distribution of the species, but this is valid up to a specific temperature in the near future. Parmoon et al. [52] calls it the optimum temperature, since a small increase affects the germination of the species, contracts the area of distribution, and reduces the ecological niche. According to Gómez-Días et al. [25], an increase above the optimum temperature affects the hormonal mechanisms involved in flowering, fruiting, and germination of the species, which is reflected in the decrease of surface suitable for germination and/or growth. Changes in distribution and phenology are expected with increasing temperature [7]. Temperature affects the hormonal mechanisms involved in the flowering and fruit set, which impacts the dispersal and distribution of species [46].

Particularly, in relation to flowering, it is known that this is determined by temperature and photoperiod [2]. It is a stage that is sensitive to photothermal changes; therefore, a change in the average temperature can modify the synchrony with pollinators and rainy periods. In this sense, climate change could promote earlier flowering, as temperatures continue to rise [3]. This will impact seed production and all associated phenological processes [1]. Additionally, the production, dispersal, and germination of seeds not only respond to instantaneous thermal signals, but also incorporate information from the thermal history of the plant’s ancestors (thermal memory) [19]. 

Since seeds are the dispersal unit of plants, germination determines the regeneration site, which affects the assembly of species in plant communities. Therefore, thermal niche breadth of germination is related to species distribution across different ecosystems, delimiting the regeneration site [53]. According to Fernández-Pascual et al. [19], models of potential distribution should consider how the future climate will affect germination percentage and velocity, as these two traits are important for the individuals’ dispersibility of populations as well as the vigor of their seedlings.

In general, there was an expansion of the potential distribution in most of the states studied for both the conservative and drastic scenario for 2050 and 2070. These results are in agreement with those found in Hernández-Ramos et al. [26], showing that *C. odorata* has a high probability of being distributed in the southern Yucatan Peninsula, north and south of Chiapas, and in the Gulf coastal plain, which is reflected in our study. On the other hand, the states of Chihuahua, Mexico, Morelos, Guerrero, and Durango suffer a contraction of the potential distribution for both the conservative and drastic scenarios. The most extreme case will be the state of Chihuahua, which will lose environmental conditions for the distribution of the species in the distant future. Therefore, it is recommended that the populations of the species in the state are monitored and the collection of seeds to conserve local genotypes is carried out. Similarly, other studies predict a 0.2% decrease in the potential distribution for the state of Hidalgo under the English model (HadCM3 phase 3 of the IPCC) by 2050 [25]. At the same time, Garza-López et al. [45] predicted a decrease in the distribution of *S. macrophylla* in Campeche, Chiapas, and Yucatán as well as complete disappearance in Quintana Roo by 2030.

On the other hand, the models generated indicate that the state of Guanajuato in the future will present favorable climatic conditions for the species, of which there is no record in previous studies. Therefore, it is recommended that provenance genetic tests are carried out and the genotypes that show the greatest adaptation are selected. With the increase in temperature, Michoacan and Hidalgo’s states will present a positive trend in surface for the distribution of species. This can be explained by the temperature rise from 1.9 to 4.7 °C, which the NCAR model (CCSM4) predicts in the far and intermediate future, which will favor the ideal conditions for the potential distribution of *C. odorata*. This coincides with Ortiz-Solorio [54], who mentioned that many physiological processes of trees are influenced by temperature. Therefore, increases in this bioclimatic element will accelerate the hormonal mechanisms of flowering and fruiting, which favors seed dispersal and impacts the distribution of the species.

According to the American model (CCSM4), an increase in temperature, a decrease in rainfall, and marked drought events are expected, altering the climatic conditions of forest regions, making them warmer and drier. Therefore, changes are expected in the five most important biomes for Mexico [47]. Tropical rain forests, mountain rain forests, and temperate forests are expected to contract dramatically. In contrast, the seasonal dry tropical forest and xerophilous scrub will expand its distribution considerably. Contrary to what would be expected, these climatic changes will favor *C. odorata*, according to the distribution model. Based on the range of optimal germination temperatures, conditions with suitable climatic habitat for the species are expected until 2070, which coincides with that which was reported for another tropical species *S. macrophylla* [46]. However, Fremout et al. [55] mentioned that tropical forests face greater threats from climate change, such as change in land use, overexploitation of forests, and overgrazing—activities that have reduced two-thirds of the original coverage in the Americas. Another serious threat is pests, which, with the increase in temperature, are expected to shorten their life cycle and can cause further damage. Therefore, the species under study is highly vulnerable to all these threats.

This study is the first to report the potential distribution of *C. odorata* based on its thermal niche for seed germination. The approach developed here constitutes an alternative methodology for estimating the potential distribution of species, considering the optimum germination temperature range, which is the most critical stage in the establishment and development of plant species. The results of this study complement, in ecophysiological terms, the conventional methods of studies of potential distribution and allow for the visualization of the perspectives including physiological, ecophysiological, and phenological components in the bioclimatic variables that are conventionally used to define the distribution niches of the species.

## 4. Materials and Methods

### 4.1. Study Species

*Cedrela odorata* L. (Meliaceae) is a tree native to tropical America (Figure 4) [56], distributed from northern Mexico to northern Argentina, up to 1200 m a.s.l. [25]. It is found in the humid and subhumid tropics and grows in association with tropical deciduous, sub-deciduous, and rain forests, as well as tropical montane cloud forest [57]. In Mexico, it is distributed along the Gulf slope, from southern Tamaulipas and southeastern San Luis Potosí to the Yucatan Peninsula. On the Pacific slope, it is distributed from southern Sonora to southern Guerrero, the Central Depression, and the coast of Chiapas [58], where it can be found in warm and semi-warm climates, and it requires soil that is fertile with good drainage [57].

### 4.2. Study Area

The study area was defined for Mexico according to the database of the National Inventory of Forests and Soils 2009–2013, in which the distribution of *C. odorata* was delimited with the humid and subhumid tropics and grows in association with tropical deciduous forest, sub-deciduous, and evergreen or rain forests, as well as tropical montane cloud forest (Figure 5) [20,58].

### 4.3. Data Acquisition

A georeferenced database of *C. odorata* was constructed from the presence data of the species collected in the country using the information contained in the Global Biodiversity Information Facility (GBIF) platform of copies deposited in herbaria around the world (https://www.gbif.org/ accesed on 18 November 2021) [59]. The database was supplemented with records obtained in field collections by personnel from the Seed Bank of the Iztacala Faculty of Higher Studies, UNAM. The database was filtered to eliminate data that were incomplete, as well as when located in urban areas, roadways, rivers, and agricultural areas, for which the land use map was used [60]. For the construction of the distribution models of *C. odorata*, a special filtering of the records of the presence of the species was carried out, which consisted of selecting only the records that met the condition of being in the optimal range of cardinal temperatures for germination, determined for the species in a previous work [7]. According to the nearest neighbor method in ArcMap 10.3 and at the spatial resolution of the layers in vector format, the minimum spatial distance between occurrence records was 5 km [61]. 

### 4.4. Climate Variables

Nineteen environmental variables were used at a spatial resolution of 30 arc seconds, of which eleven were temperature variables. These data were obtained from the BioClim database (Table 2) from the period of 1970 to 2000 for the current distribution [62], at a resolution of 1 km^2^ per pixel [61]. The layers, in vector format (INEGI, VI series layer) of land use and vegetation, were taken from CONABIO [63].

### 4.5. Distribution Model and Optimum Germination Temperature Range

To model the distribution of *C. odorata* according to the cardinal temperatures for seed germination, we used the values previously identified in a previous study by Sampayo-Maldonado et al. [7]. These authors determined a minimum temperature (Tb) of −0.5 ± 0.09 °C, an optimum temperature (To) of 38 ± 1.6 °C, and a maximum temperature (Tc) of 53.3 ± 2.1 °C for this species [7]. To define the optimum germination temperature range, the normal behavior assumption of germination values was presumed. In this sense, considering the mean value of To ± 2 standard deviations, the optimal range of 34.8 to 41.2 °C was defined, which included 95% of the population in this parameter. To model the distribution of *C. odorata*, the collection sites that were located in the zones that meet the optimum germination temperature range were selected.

### 4.6. Modeling Current Distributions of C. odorata

Species distribution models (SDM) were built with MaxEnt (software version 3.4.1^®^) (Steven J. Phillips, Miroslav Dudik and Robert E. Schapire. Nueva York, NY, USA) to carry out the analyses according to the collection sites and distribution of *C. odorata* based on the maximum entropy model. The algorithm estimates the probable distribution of species and uses a set of environmental variables to determine the maximum entropy distribution. High predictive precision is achieved, in a logistical format, by improving the calibration of the model, which provides greater representativeness of suitability [64]. In the analysis, 1373 georeferenced sites with the presence of populations of the species were used. Of the sites with the presence of the species, 256 were used and were those that were distributed in the optimum germination temperature range (34.8 to 41.2 °C). These numbers of sites was considered a sufficient sample size [65], so the prediction of the geographic distribution for the species was acceptable.

To generate the current distribution of optimal climate habitat for *C. odorata*, BioClim version 2.0 (Richard G. Pearson. MD, USA) [62] http://www.worldclim.org accessed on 18 November 2021 was used, with data from the period 1970 to 2000. The distribution of optimal climate habitat for the species under the current climate was modeled using the MaxEnt algorithm. This process is known as environmental niche modeling [41,66] or climate niche modeling [45]. Additionally, the jackknife method was used, implemented in MaxEnt, to indicate the relative contribution of each environment al variable in the model. With the environmental variables that contributed the most to the model, the program was executed again to generate the maps. The spatial distribution of the optimal climate habitat was obtained using the tool ArcMap 10.3^®^, obtaining the number of pixels and transforming to km^2^. Finally, the maps of the distribution of the species were generated for the current climate, with a probability greater than 20% of identifying areas where the temperature will have positive effects on seed germination in the understory.

To overcome the limitations of the distribution models of the species in this study, the Pearson correlation coefficient was used to test the multicollinearity among the 21 environmental variables. The selection of predictive variables was based on the following criteria [64,65,66,67]: those with the greatest contribution according to the jackknife test, uncorrelated (r ≤ 0.8), with normal distribution, and by the type of response of the environmental variable, in addition to taking into account previous work on the distribution of the species, eliminating areas taking into account barriers that make the dispersal of the species impossible, restricting the results by biogeographical regions, and using physiological data on the germination of the species. 

To validate the model, the recommendations of Peterson and Soberón [67] were followed. The test was carried out using 30% of the data, which were separated randomly from the total localities with presence of the species. The goodness-of-fit of the predictions of the model were evaluated using the area under the curve (AUC) and the receiver operating characteristics (ROC). The results were expressed as a ratio of the ROC curve observed to a random curve, where both were truncated to the area delimited by the error threshold. The model obtained, using 70% of the data, was reclassified to values between 0 and 1, where 0 is less ideal (lower probability), and 1 is the most ideal (higher probability) for the presence of the species. The jackknife procedure was used to reduce the number of environmental variables so that only the variables that had the highest contribution to the model prediction are shown. In addition, the generated models were evaluated through the omission rate, Aikaike information criteria, or model complexity (AIC) [67]. Following the parsimony principle, we discard the most complex models and choose the simplest model, the one with the lowest (AIC), according to the Aikaike information criterion.

### 4.7. Model of Potential Distribution of C. odorata

Climate change scenarios were estimated on the basis of the difference between current and future climate layers according to the United States National Center for Atmospheric Research (NCAR) model (CCSM4). For this, the General Circulation Model (GCM) climate layers for CCSM4 were downloaded, which were generated on the basis of Regional Models from the Coupled Model Intercomparison Project, Phase 5 (CMIP 5), of the Intergovernmental Panel on Climate Change (IPCC), projected for a horizon of intermediate future 2050 (average for 2041–2060), and for 2070, as a horizon of far future (average of 2061–2080), with two contrasting representative concentration pathways of Watts/m^2^ (RCP2.6 and RCP8.5) representing two extreme scenarios, with the lowest and the highest CO_2_ emissions classified as conservative and drastic scenarios, respectively [68]. For the intermediate future (2050), it predicts average annual temperature increase for RCP2.6 of 1.9 and of 3.8 °C for the RCP8.5, while for the far future horizon (2070), it predicts 2.7 and 4.7 °C for the RCP2.6 and RCP8.5, respectively, of the average annual temperature increase [69].

The potential future distribution was projected using the MaxEnt algorithm (SDM), using the geographic records and the intermediate (2050) and far (2070) future temperature grids. Firstly, this tool models optimal climate habitat under the current climate. Once that model has been constructed, the projection is repeated, but this time using the model with the grids of climate variables for the decades centered on 2050 and 2070, estimated using the NCAR model (CCSM4) with an RCP2.6 and RCP8.5 Watts/m^2^ (lowest and the highest CO_2_ emissions), classified as conservative and drastic models, respectively. Finally, maps of the distribution of the species in future climate were generated, with a probability greater than 20%.

## 5. Conclusions

In general, due to climate change, the potential distribution of *C. odorata* will expand in most of the states of the country with some exceptions (Chihuahua, Mexico, Morelos, Guerrero, and Durango). The most extreme case will be the state of Chihuahua, which will lose optimal conditions for the distribution of the species. With the increase in temperature, the appearance of adequate environmental conditions for the species distribution in the state of Guanajuato is expected. From the thermal variables, the coldest month and the annual temperature oscillation are those that contributed the most to the prediction models. Therefore, on the basis or our results, it was possible to predict the potential distribution of *C. odorata* based on its thermal niche for seed germination. This will allow actions to be undertaken to monitor populations that are vulnerable to climate change, as well as activities such as seed collection to conserve local genotypes whose populations will disappear, and to carry out genetic studies of provenances in the new areas that will have adequate environmental conditions to the potential distribution to the species.

## Figures and Tables

**Figure 1 plants-12-00150-f001:**
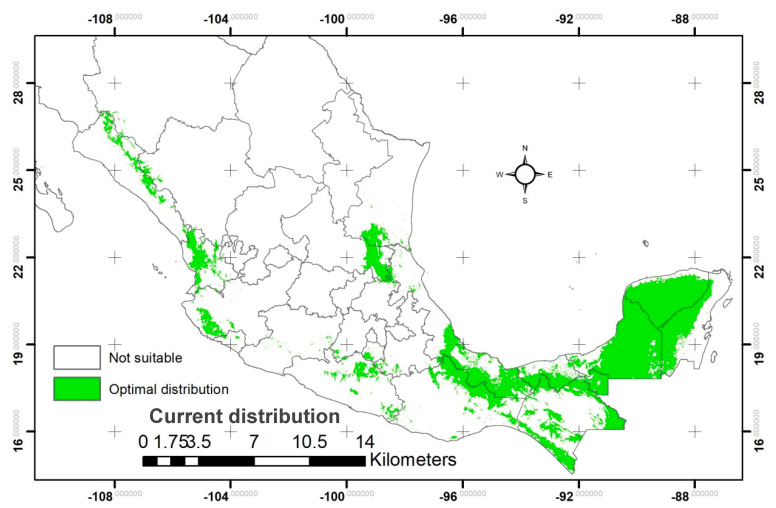
Current distribution of *C. odorata*. In green, the optimal distribution according to the optimum germination temperature range.

**Figure 2 plants-12-00150-f002:**
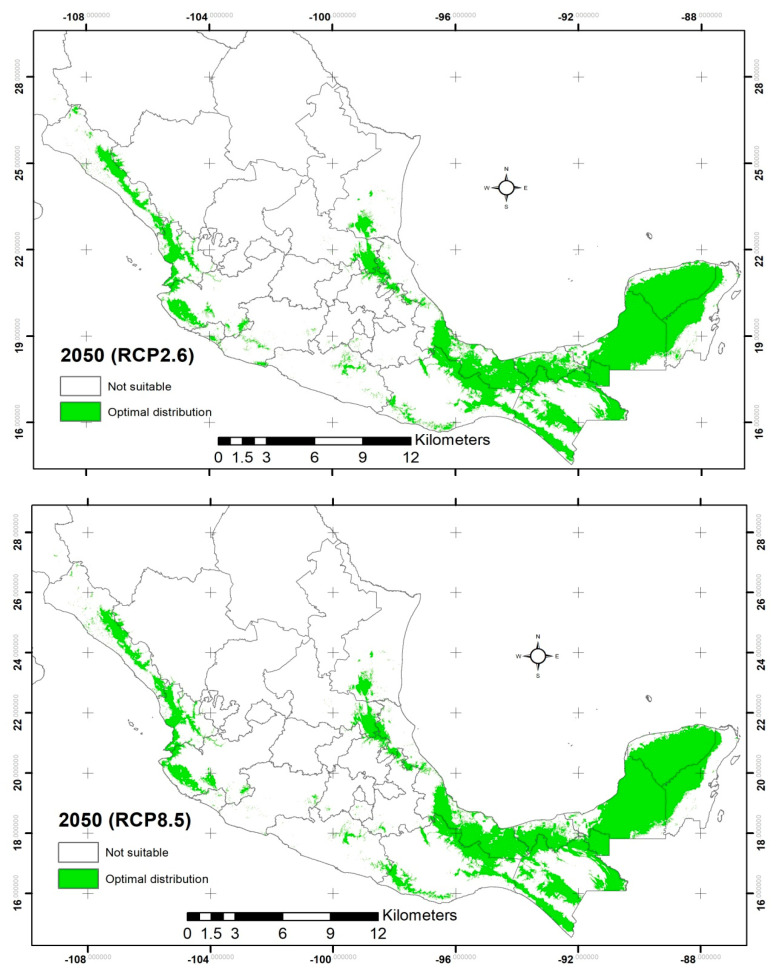
Potential distribution for the intermediate future (2050) of *C. odorata* in the NCAR model (CCSM4); for the conservative (RCP2.6) and drastic (RCP8.5) scenarios according to the optimum germination temperature range.

**Figure 3 plants-12-00150-f003:**
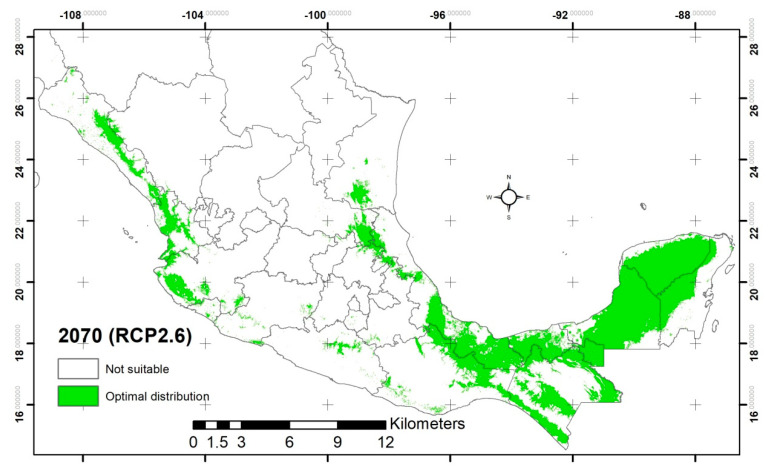
Potential distribution for the far future (2070) of *C. odorata*, in the NCAR model (CCSM4); for the conservative (RCP2.6) and drastic (RCP8.5) scenarios according to the optimal germination temperature range.

**Figure 4 plants-12-00150-f004:**
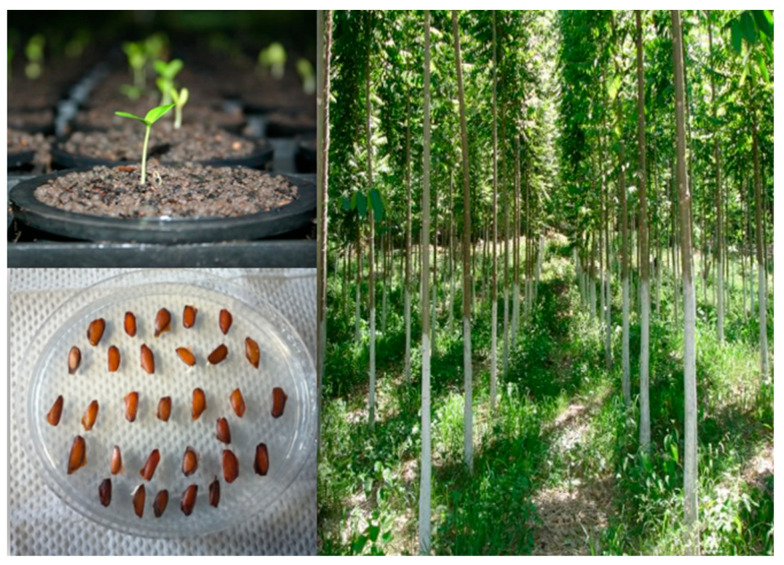
Seeds, seedlings, and young trees of *C. odorata*.

**Figure 5 plants-12-00150-f005:**
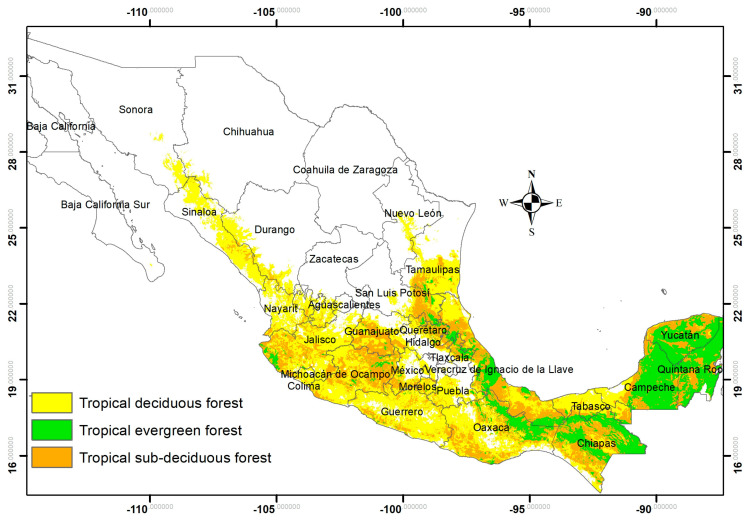
Distribution of the tropical forest in Mexico, as area of study for the distribution of *C. odorata*.

**Table 1 plants-12-00150-t001:** Surface with potential distribution of *C. odorata* according to climate change scenarios by state in the model (CCSM4).

State	Current (km^2^)	2050 (km^2^)		2070 (km^2^)	
		RCP2.6	RCP8.5	RCP2.6	RCP8.5
Campeche	49,206.37	46,696.03	46,596.33	45,717.05	46,299.80
Chiapas	26,474.27	31,882.02	32,529.20	32,424.94	33,982.65
Chihuahua	433.45	28.61	2.23	16.59	-
Colima	335.29	572.07	208.92	475.87	929.28
Durango	1508.06	1128.22	773.41	1101.02	759.43
Guanajuato	-	15.38	6.64	65.10	2.70
Guerrero	6033.25	2138.62	2268.99	2150.80	2684.70
Hidalgo	668.78	1867.85	1566.70	1831.65	1755.27
Jalisco	5044.10	8603.37	8863.37	9131.17	8990.25
Mexico	433.54	67.76	80.22	46.47	4.41
Michoacan	435.00	1391.38	409.87	1,301.65	798.17
Morelos	802.23	122.15	70.71	62.82	72.15
Nayarit	8899.30	8895.37	9016.18	9022.26	8872.49
Oaxaca	16,410.20	20,896.92	20,852.32	18,317.95	19,579.65
Puebla	1341.35	848.88	1368.41	1309.98	1862.81
Queretaro	71.61	137.77	96.53	158.12	65.26
Quintana Roo	20,136.19	17,307.75	16,330.78	15,596.01	18,351.02
San Luis Potosi	6848.31	4888.47	5112.18	4445.60	5200.63
Sinaloa	6586.40	9883.49	9728.92	9754.78	9823.81
Sonora	16.45	11.42	32.58	3.78	31.98
Tabasco	14,217.66	17,776.39	16,326.25	18,410.85	16,609.55
Tamaulipas	4623.80	3757.83	3467.14	3779.01	3526.20
Veracruz	24,716.46	31,145.75	31,821.38	31,705.65	32,537.99
Yucatan	33,538.96	33,827.35	34,003.95	33,109.55	33,941.25
Suitability area (km^2^)	228,780.66	243,890.85	241,533.21	239,938.67	246,681.45
Increment area (km^2^)	N/A	15,110.19	12,752.55	11,158.01	17,900.79
Percentage of increment (%)	N/A	6.60	5.57	4.87	7.82

**Table 2 plants-12-00150-t002:** Environmental variables used in this study.

Abbreviation	Environmental Variable	Units	Used for Modeling
Bio1	Annual average temperature	(°C)	No
Bio2	Diurnal temperature oscillation	(°C)	Yes
Bio3	Isothermality (Bio2/Bio7) × 100	(°C)	No
Bio4	Temperature seasonality (standard deviation × 100)	(°C)	Yes
Bio5	Maximum average temperature of the warmest period	(°C)	No
Bio6	Minimum temperature of the coldest month	(°C)	Yes
Bio7	Annual temperature oscillation (Bio5–Bio6)	(°C)	Yes
Bio8	Average temperature of the wettest month	(°C)	No
Bio9	Average temperature of the driest month	(°C)	Yes
Bio10	Average temperature of the warmest quarter	(°C)	No
Bio11	Average temperature of the coldest quarter	(°C)	No
Bio12	Annual precipitation	(mm)	Yes
Bio13	Precipitation during the wettest period	(mm)	No
Bio14	Precipitation during the driest period	(mm)	Yes
Bio15	Precipitation seasonality (coefficient of variation)	CV	No
Bio16	Precipitation during the wettest trimester	(mm)	No
Bio17	Precipitation during the driest trimester	(mm)	No
Bio18	Precipitation during the warmest trimester	(mm)	No
Bio19	Precipitation during the coldest trimester	(mm)	No
Altitude	Elevation	(m)	No
Soils	Soil type	-	Yes

Source: WorldClim [62]; CONABIO [63].

## Data Availability

Geographic coordinate data of collected species and environmental variables were described in Section 4.2, Section 4.3 and Section 4.4 of the article.

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
