# Peer review of "Potential Distribution of *Cedrela odorata* L. in Mexico according to Its Optimal Thermal Range for Seed Germination under Different Climate Change Scenarios"

_plants, 2022, doi:10.3390/plants12010150_

Round 1
Reviewer 1 Report
Please see attached.

Reviewer 2 Report
Dear colleagues!
In their study, the authors proposed a prognostic model of the development of the population of Cedrela odorata L. in Mexico during climate change. In their model, the authors laid down climatic indicators that would have an impact on seed germination. The results of the work are of fundamental importance for the population biology of plants. In the applied aspect, the authors' research will help predict the productivity of the population of this economically valuable plant species.
There is competent and persuasive introduction. The authors also described in detail the results of the work, illustrated their judgments, and conducted a competent discussion of the results.
I ask the authors to take into account my recommendations and answer a few questions:
1. Please provide a photo of the object of study.
2. Specify the status of the species in the environmental aspect.
3. Is there any data on the distribution area of Cedrela odorata in Mexico in the past years?
4. Which factor makes the greatest contribution to the change in the population of Cedrela odorata and the change in the distribution area at the present time – human economic activity or climatic changes?
5. Climate change will also affect populations of the pest species Cedrela odorata. I assume that it will cause an increase in their numbers. What is the impact of insect pests on the wood of Cedrela odorata at the present time? What are your forecasts for the development of the relationship "Cedrela odorata – insect pests" in the future?
Reviewer 3 Report
Accept.
Author Response
Thank you very much for your favorable opinion of this manuscript.
Reviewer 4 Report
The Authors presented a very interesting study: The analysis of the possible future territorial distribution of a plant species, also important from an economic point of view, as a result of climate change. The climate challenge is becoming more and more pressing and it is therefore essential to develop models that help predict how to deal with it preserving biological diversity and different crops. In this paper the Authors have carried out an in-depth study and all section of the paper are well written. The results are presented in a clear way and offer many food for thought. Their methodological approach is appealing and deserves to be taken into consideration. In addition their conclusion are significative to monitor the anthropogenic activities and impact and to conserve the various plant genotypes. In my opinion this paper deserves the publication
Author Response
We appreciate your comments on the subject of study and your favorable opinion of this manuscript.
Round 2
Reviewer 1 Report
The authors have addressed the majority of my concerns. However, I have still concerns as far as the use of ‘SSP’ and ‘RCP’ are concerned. The authors have changed ‘RCP’ for ‘SSP’ throughout the manuscript; please these two represent two different datasets. Unless the authors have repeated the analysis? The RCP should be used instead of SSP.
Author Response
Following the reviewer's suggestion, RCP is indicated instead of SSP as the database for the scenarios studied in the prediction of the distribution of C. odorata. The results obtained, their analysis and discussion, are derived from this model.
We acknowledge the relevance of the reviewer's suggestion and appreciate his contribution in improving this manuscript.